# Quantitative Multiplexed Proteomics Could Assist Therapeutic Decision Making in Non-Small Cell Lung Cancer Patients with Ambiguous ALK Test Results

**DOI:** 10.3390/cancers13102337

**Published:** 2021-05-12

**Authors:** Ho Jung An, Eunkyung An, Shahrooz Rabizadeh, Wei-Li Liao, Jon Burrows, Todd Hembrough, Jin Hyung Kang, Chan Kwon Park, Tae-Jung Kim

**Affiliations:** 1Department of Medical Oncology, St. Vincent’s Hospital, College of Medicine, The Catholic University of Korea, Seoul 06591, Korea; meicy@catholic.ac.kr; 2NantOmics, Culver City, CA 90232, USA; Eunkyung.An@nantomics.com (E.A.); Shahrooz@nantworks.com (S.R.); w.liao@oncoplexdx.com (W.-L.L.); 3OncoPlex Diagnostics, Rockville, MD 20850, USA; jon.burrows@oncoplexdx.com (J.B.); todd.hembrough@nantomics.com (T.H.); 4Department of Medical Oncology, Seoul St. Mary’s Hospital, College of Medicine, The Catholic University of Korea, Seoul 06591, Korea; jinkang@catholic.ac.kr; 5Division of Pulmonology, Department of Internal Medicine, Yeouido St. Mary’s Hospital, College of Medicine, The Catholic University of Korea, Seoul 06591, Korea; ckpaul@catholic.ac.kr; 6Department of Hospital Pathology, Yeouido St. Mary’s Hospital, College of Medicine, The Catholic University of Korea, Seoul 06591, Korea

**Keywords:** ALK, non-small-cell lung cancer, proteomics, selected reaction monitoring, fluorescent in situ hybridization, immunohistochemistry, tyrosine kinase inhibitor, chemotherapy

## Abstract

**Simple Summary:**

Therapeutic guidance in non-small cell lung cancer (NSCLC) patients with discordant anaplastic lymphoma kinase (*ALK*) fluorescent in situ hybridization (+) where immunohistochemistry (IHC) (−) results are challenging. Selected reaction monitoring (SRM) quantitative multiplexed proteomics could detect ALK protein in NSCLC samples with delayed fixation where a conventional IHC method failed. ALK protein detection by the SRM method was associated with good responses on ALK inhibitors. It also could detect various predictive proteins for conventional chemotherapy at the same time, and combined results were related to clinical outcomes in this population. The SRM may provide additional information for therapeutic decision making in NSCLC patients with ambiguous ALK test results.

**Abstract:**

Therapeutic guidance in non-small cell lung cancer (NSCLC) tumors that are positive for anaplastic lymphoma kinase (*ALK*) fluorescent in situ hybridization (FISH), but negative for ALK immunohistochemistry, is still challenging. Parallel routine screening of 4588 NSCLC cases identified 22 discordant cases. We rechecked these samples using ALK antibodies and selected reaction monitoring (SRM) quantitative multiplexed proteomics screening multiple protein targets, including ALK and MET for the ALK tyrosine kinase inhibitor (TKI), and FR-alpha, hENT1, RRM1, TUBB3, ERCC1, and XRCC1 for chemotherapy. The presence of ALK (31.8%), MET (36.4%), FR-alpha (72.7%), hENT1 (18.2%), RRM1 (31.8%), TUBB3 (72.9%), ERCC1 (4.5%), and a low level of XRCC1 (54.4%) correlated with clinical outcomes. SRM was more sensitive than the ALK D5F3 assay. Among the eight cases receiving ALK TKI, four cases with ALK or MET detected by SRM had complete or partial responses, whereas four cases without ALK or MET showed progression. Twenty-seven treatment outcomes from 20 cases were assessed and cases expressing more than half of the specific predictive proteins were sensitive to matching therapeutic agents and showed longer progression-free survival than the other cases (*p* < 0.001). SRM showed a potential role in therapeutic decision making in NSCLC patients with ambiguous ALK test results.

## 1. Introduction

Precise diagnosis and classification of lung cancers are essential for selecting the appropriate therapeutics. Recent advances in targeted therapies for non-small cell lung cancer (NSCLC) have been achieved largely due to developments in detecting specific pathogenic mutations [1]. For example, patients with NSCLC with an anaplastic lymphoma kinase (*ALK*) gene rearrangement are more responsive to an ALK tyrosine kinase inhibitor (TKI), such as crizotinib, than to chemotherapy [2]. The *ALK* gene rearrangement is observed in 3–5% of adenocarcinomas of the lung [2,3], and, hence, identifying patients with ALK aberrations who might benefit from the kinase inhibitor treatment is critical for effective NSCLC treatment. Fluorescent in situ hybridization (FISH) has been the standard method for detecting *ALK* rearrangements and ALK immunohistochemistry (IHC) has shown high correlation with it [4,5]. At present, the *ALK* break-apart FISH and the Ventana D5F3 ALK assay represent the companion diagnostic methods approved by the U.S. Food and Drug Administration to detect ALK aberrations [6]. On rare occasions, *ALK* FISH and ALK IHC can yield discordant results. This ambiguity may be explained by a false positive *ALK* FISH result, absence of the ALK protein, non-specific IHC staining, limited antibody sensitivity, or failure to detect gene rearrangements with minimal separation or atypical signals [7]. Many discordant cases have been associated with a borderline *ALK* rearrangement or copy number gain [8]. In practice, not all patients with *ALK* FISH positivity have benefited from ALK TKI treatment, whereas some *ALK* FISH-negative/ALK IHC-positive patients have responded to ALK TKI [9,10]. Furthermore, therapeutic strategies for use in patients with discordant or ambiguous ALK status are limited. These discrepancies across the standard ALK testing methods have fueled a debate regarding selection of patients best suitable for ALK TKI treatment, pointing toward a need for improved ALK diagnostic methods. Selected reaction monitoring (SRM) is a mass spectrometry-based proteomic method for quantitatively assessing predetermined candidate biomarkers in multiple samples in a reproducible and quantitatively precise manner [11]. Therefore, in the present study, we applied SRM to detect ALK and other protein biomarkers in patients with NSCLC with borderline or positive *ALK* FISH outcomes but negative ALK IHC results. SRM could potentially guide therapeutic decision making for patients with NSCLC with discordant *ALK* FISH and ALK IHC outcomes.

## 2. Results

### 2.1. Patient Characteristics

The baseline characteristics of the 22 patients with discordant ALK testing results are shown in Table 1. The median age was 57 years (range, 25–90 years), 13 cases (59.1%) were male, and 14 cases (63.6%) were non-smokers. The majority of patients showed metastatic or recurrent disease at presentation (72.8%) and adenocarcinoma (91.0%). Fifteen samples (68.2%) were obtained from biopsies from multiple sites, and seven (31.8%) samples were obtained from surgical specimens. There were eleven cases (50.0%) with direct formalin fixation (<30 min), four cases (18.2%) with direct but delayed fixation (>30 min), and seven cases (31.8%) with fixation after overnight refrigeration. Eight cases were positive for *ALK* FISH (gene split, >20% tumor cells), and 14 cases were borderline-positive for *ALK* FISH (gene split, 15–20% tumor cells).

### 2.2. Comparison of ALK FISH, ALK IHC, and SRM Assay

The results of *ALK* FISH, ALK IHC, and SRM of each patient are summarized in Appendix A. Detection using the 5A4 antibody revealed four new positive cases (two 3+, one 2+, and one 1+), while the D5F3 assay identified the four 5A4-positive cases and an additional case (four diffuse positive and one focal strong positive (5A4 negative)). The sample of the case negative for 5A4 but positive for D5F3 was a biopsy of the brain metastatic tumor with direct fixation. Although five cases were positive according to the additional sensitive IHC, SRM detected seven ALK protein-positive cases, including the previously mentioned five cases detected by IHC. Two out of the seven ALK-positive cases identified by SRM alone were clearly negative by IHC with a score 0 for 5A4 and weak negative staining for D5F3. Pre-analytical data for SRM-positive, IHC-negative cases revealed that one (case 12) sample was a lobectomy with delayed fixation, while the other (case 14) was a biopsy of the brain metastatic tumor that underwent overnight refrigeration before fixation. Among the eight cases with positive *ALK* FISH (>20% tumor cells), IHC detected four positive cases. However, SRM detected six positive cases. Among the six borderline *ALK* FISH-positive cases, IHC and SRM each identified a positive case (Table 2). Figure 1 shows the representative FISH, IHC, and SRM results. The results of *ALK* FISH, ALK IHC, and SRM were compared with respect to the fixation status. A positive *ALK* FISH (>20% tumor cells), positive IHC, and positive SRM were significantly associated with direct fixation (*p* < 0.05) (Table 3). The ALK protein was not detected using IHC in specimens with delayed fixation, while SRM could detect the ALK protein in these cases (case numbers 12 and 14, Appendix A).

### 2.3. Protein Quantitation by SRM and the Associated Clinical Outcomes

In addition to the ALK protein, MET (hepatocyte growth factor receptor), another target of crizotinib, and additional protein biomarkers for chemotherapeutic agents were quantified by multiplexed SRM (Figure 2, Appendix A). MET was detected in eight patients (36.4%), FR-alpha (folate receptor alpha, a marker for pemetrexed sensitivity) in 16 patients (72.7%), hENT1 (human equilibrative nucleoside transporter 1, a marker for gemcitabine sensitivity) in four patients (18.2%), RRM1 (ribonucleotide-diphosphate reductase M1, a resistance marker for gemcitabine sensitivity) in seven patients (31.8%), TUBB3 (tubulin beta-3 chain, a resistance marker for anti-tubulin agents) in 16 patients (72.8%), ERCC1 (excision repair cross-complementation group 1, a marker for platinum sensitivity) in one patient, and low levels of XRCC1 (X-ray repair cross-complementing protein 1, a resistance marker for platinum sensitivity) were observed in 12 patients (54.5%). The XRCC1 protein level was considered low or high based on its median value of 479.4 amol/ug. Multiplexed SRM revealed that ALK and MET were co-expressed in less than 50% of the cases, and MET was detected in four out of thirteen borderline-positive *ALK* FISH samples. We analyzed each treatment response in relation to the protein levels (detected vs. not detected or low vs. high). Of the eight crizotinib-treated cases (five strongly positive and three borderline-positive for *ALK* FISH), two cases (case numbers 14 and 19, Appendix A) with detectable levels of both ALK and MET proteins had a complete response including 13.8 months of PFS and a partial response (PR) including 46.3 months of PFS, respectively. Moreover, two cases that were detected with either the ALK or MET protein (case numbers 11 and 21, Appendix A) had PRs. However, four cases (case numbers 5, 7, 18, and 20, Appendix A) that did not exhibit detectable ALK or MET proteins via SRM did not respond to ALK TKI treatment, even though their *ALK* FISH results were positive (one fully positive and three borderline-positive). Among the patients (*n* = 2) who received LDK378, which is a second-generation ALK inhibitor, a case with the ALK protein detected by SRM had a PR (case number 21, Appendix A), and another case without a detectable *ALK* protein developed progressive disease (case number 18, Appendix A). The ALK protein was detected by SRM in one out of thirteen borderline-positive *ALK* FISH cases, which is less frequent than in *ALK* FISH-positive cases (six out of eight *ALK* FISH-positive cases, Appendix A). For each combinational chemotherapeutic treatment, the treatments administered in each patient before a treatment change due to progression were categorized into one of the five SRM marker groups. For example, case number 10 received gemcitabine/cisplatin. Hence, the predictive biomarkers were RRM1 and hENT1 for gemcitabine, and ERCC1 and XRCC1 for platinum. Case number 10 expressed no RRM1 (beneficial), no hENT1 (not beneficial), no ERCC1 (beneficial), and XRCC1 (not beneficial) higher than the median 479.4 amol/ug. Thus, the score was determined to be two out of four (50.0%) and case number 10 was classified into group 3. After excluding the cases receiving an adjuvant (case numbers 3 and 4) or no treatment (case numbers 16 and 17), clinical responses were evaluated for 25 treatment regimens in 18 cases. There were 5, 1, 1, 8, and 10 responses allotted to SRM marker groups 0, 1, 2, 3, and 4, respectively. All patients in groups 0, 1, and 2 experienced progressive disease with ongoing treatment (Appendix A). In groups 3 and 4 (*n* = 18), 12 (66.7%) cases had PR and one (5.6%) had a complete response (Figure 3A) to treatment. The PFS was significantly longer in the high-scoring groups (groups 3 and 4) than in the low-scoring groups (groups 0–2), according to the Kaplan-Meier survival analysis (*p* < 0.001) (Figure 3B).

## 3. Discussion

In routine clinical settings, the discordance between ALK FISH and ALK IHC results is rarely observed, which is consistent with our previous study [12]. Combined use of ALK IHC and ALK FISH improves the detection of ALK aberrations [13]. The participating hospitals and we have instituted a diagnostic algorithm that combines ALK FISH and ALK IHC in routine screening. In the present study, we collected samples with positive or borderline *ALK* FISH and negative ALK IHC for re-evaluation using IHC with antibodies that are more sensitive and using a multiplexed SRM assay. The IHC results for the ALK protein were inconsistent, possibly due to the type of antibody and fixation conditions used, which may have influenced the ALK IHC results. The VENTANA D5F3 companion diagnostic assay showed a weak staining-negative IHC in two *ALK* FISH (+) and SRM (+) cases, wherein the samples underwent delayed fixation. Improper fixation can influence IHC sensitivity [14]. SRM is independent of the protein epitope status and showed superior sensitivity in detecting the *ALK* protein in samples with poor fixation in our cohort. SRM showed superior sensitivity than IHC in samples from lobectomies, which tended to be poorly fixed and had longer ischemic time than the biopsied samples [15]. The sensitivity and specificity of ALK IHC have been previously reported, as 67–100% and 95–99%, respectively [3,4]. Vital discordance between IHC and FISH results has been reported in the initial period of routine *ALK* screening [7,8,16,17]. In NSCLC with *ALK* rearrangements, the ALK protein is expressed at levels much lower than that in lymphomas with *ALK* rearrangements [18]. A low *ALK* gene copy number and borderline *ALK* rearrangement are the alternatively proposed mechanisms for the low ALK protein level in NSCLC [8,19]. FISH would work better with some breakpoint variants than the other methods [20]. ALK protein expression may be caused by not only *ALK* gene rearrangements, but other mechanisms such as *ALK* amplification. For example, a copy number gain and point-activating mutations in *ALK* have been observed in tumors with ALK protein expression [21,22]. For the ALK TKI treatment, *ALK* FISH need not always be concordant with ALK IHC. However, it could be crucial to consider the possibility of false positivity or false negativity, due to poor sample processing, for selecting patients best suitable for ALK TKI treatment in routine practice. A comparative study of *ALK* detection by FISH, IHC, and next-generation sequencing showed a false-negative rate in FISH [23]. Since we considered *ALK* FISH as the gold standard, our cohort did not exhibit a false-negative rate. In the present study, SRM assays detected the ALK protein in seven discordant cases (six out of eight fully positive and one out of 13 borderline-positive for *ALK* FISH). Case numbers 11 and 20 were fully positive for *ALK* FISH, but the ALK protein was not detected by either IHC or SRM. In such scenarios, co-existing complex patterns (deleted, split, and amplified/polysomic) of the *ALK* gene have been observed by FISH analysis [10]. Recent reports suggest that MET is frequently expressed in NSCLC with *ALK* rearrangements [8,24]. Three out of six ALK-positive cases co-expressed both ALK and MET according to SRM. Moreover, one of the two cases fully positive for *ALK* FISH expressed MET. MET amplification without *ALK* rearrangement is known to predict responses to crizotinib [25,26]. A positive response to crizotinib has also been reported in tumors borderline-positive for *ALK* FISH with a high MET expression [8]. In the present study, five out of eight MET-positive samples showed a borderline-positive *ALK* rearrangement. Among these five cases, one (case number 11) received crizotinib treatment and experienced a PR with 19.8 months PFS. Multiplexed SRM had a higher predictive value for the ALK TKI treatment response than other methods. Hence, further clinical studies with larger cohorts are necessary to determine if ALK protein detection by SRM is predictive of a response to ALK inhibitors. An important advantage of SRM is its ability to quantify multiple drug-targeted markers in small amounts of FFPE tissues and in poorly fixed samples, which is challenging for IHC-based testing [5]. Furthermore, SRM has superior multiplexing capabilities, and can provide information about potential sensitivity to alternative chemotherapeutic options for patients who are not suitable candidates for ALK TKI treatment. In a clinical setting, consideration of tissue requirement, cost, and turnaround time in the diagnostic platform are essential [27]. The tissue requirement of the SRM assay is similar to that of FISH and a real-time polymerase chain reaction, and dissimilar to that of next generation sequencing. However, there are crucial limitations for the use of SRM in a clinical setting. First, the cost of SRM is as high as or higher than that of next generation sequencing as SRM is currently not a companion diagnostic for ALK TKI and not reimbursed. Second, the availability of SRM is extremely limited, which means the turnaround time would be longer than any other commercially available method. Our data suggest that SRM assays have superior sensitivity when compared to IHC in poorly prepared tissue specimens and might provide useful supplementary information regarding the appropriateness of treatment with ALK inhibitors or other chemotherapies. Nonetheless, the present study is a retrospective study with a small sample size and not all cohort cases had enough tissue remaining for re-evaluating with IHC and SRM, which warrants further investigation to guide therapeutic decision making for patients with NSCLC.

## 4. Materials and Methods

### 4.1. Patient Samples

From January 2012 to February 2017, we performed 4588 *ALK* FISH tests on samples from 16 participating institutions obtained as part of the routine clinical practice. We also assessed the result of *ALK* IHC performed at the individual hospitals for the 326 *ALK* rearrangement positive cases. Among these, 285 (6.2%) cases were positive for both FISH and IHC, and 41 cases (0.8%) showed a FISH-positive/IHC-negative discordant result. Of the discordant cases, 22 cases with sufficient remainder tissue samples were analyzed (Figure 4). Pre-analytical data included tissue fixation time and tissue typing. Clinical-demographic data included a pathologic diagnosis, disease stage, smoking status, performance status, and previous history of radiotherapy or surgical treatment. Overall treatment response was classified as a complete response (CR), a partial response (PR), stable disease (SD), or progressive disease (PD). Briefly, the treatment response calculated using the mean of two tumor diameters was used for classification, according to RECIST version 1.1 criteria: (1) CR, disappearance of all target lesions, (2) PR, at least a 30% reduction in the sum of the target lesion diameter, (3) SD, absence of PR or PD, (4) PD, at least a 20% increase in the sum of the target lesion diameter, or the appearance of a new lesion. Progression-free survival (PFS) for each administered drug was measured from the date of initial treatment to that of documenting progression or death. The institutional review board of the Catholic Medical Center at The Catholic University of Korea approved this study (XC14SIGI0046).

### 4.2. ALK FISH and ALK IHC

*ALK* FISH was performed using the *ALK* dual-color break-apart probe (Abbott Molecular, Abbott Park, IL, USA), according to the manufacturer’s instructions. The testing and interpretation of the results were performed as previously described [12]. Briefly, 2 μm thick FFPE tissue sections were deparaffinized, dehydrated, immersed in 0.2 N HCl, and cleaned with a wash buffer. The sections were immersed in 0.01 M citrate buffer (Abbott Molecular), boiled in a microwave for 5 min, treated with the pre-treatment reagent (Abbott Molecular) at 80 °C for 30 min, and reacted with protease and protease buffer (Abbott Molecular). After applying the probe mixture, sealed slides were incubated with Hybrite (Abbott Molecular) at 75 °C for 5 min to denature the probe, and the target DNA was sequentially incubated at 37 °C for 16 h to allow hybridization. Slides were then immersed in 0.3% NP-40 (Abbott Molecular)/2× saline sodium citrate for washing. For nuclear counterstaining, 4,6-diamidino-2-phenylindole (DAPI) II with the anti-fade compound p-phenylenediamine (Cellay, Inc., Cambridge, MA, USA) was applied. Signals for each probe were evaluated under a microscope equipped with a triple-pass filter (DAPI/Green/Orange, Abbott Molecular) and an oil immersion objective. *ALK* FISH is considered positive when: (1) a gene split is visible (as separate red and green signals) in >20% of tumor cells (fully positive) and (2) a gene split is visible in 15–20% of tumor cells (borderline-positive) [28]. The most commonly used ALK IHC probe at the participating institutions was the ALK 1 antibody CD246 (DAKO, Glostrup, Denmark). Additionally, we performed IHC staining using the ALK 5A4 antibody (Novocastra Laboratories, Ltd., Newcastle, UK) and the VENTANA ALK D5F3 assay (Ventana, Tucson, AZ, USA) for the 22 discordant cases. Detection was performed using the VENTANA BenchMark ULTRA automated slide-processing system with the OptiView DAB IHC Detection Kit (Ventana). The staining score for 5A4 was assigned as follows: 0, no staining, 1+, weak cytoplasmic staining visible under 400× magnification, 2+, moderate cytoplasmic staining clearly visible under 200× magnification, and 3+, strong cytoplasmic staining clearly visible under 40× magnification [29]. For D5F3, specimens were scored as positive if a strong granular cytoplasmic brown staining was present in tumor cells (any percentage of positive tumor cells). Homogeneous staining of all the tumor cells was not required as long as there were regions with a strong cytoplasmic staining. Cases were scored as negative if there was no cytoplasmic staining or weak cytoplasmic staining [30].

### 4.3. Multiplexed Mass Spectrometry-Based SRM

Protein targets in tumor tissues for ALK TKI and other predictive markers for chemotherapy were simultaneously quantified by mass spectrometry-based SRM, as previously described [31,32]. A tissue section was sliced for hematoxylin and eosin staining, and a minimum of two tissue sections at 10-μm thickness for at least 1 cm in diameter and consist of greater than 50% of the tumor or additional sections if the tumor is smaller than 1 cm were sliced onto the Director micro-dissection slides (OncoPlex Diagnostics, Rockville, MD, USA) and stained with eosin. Tumorous areas were marked by a pathologist on the hematoxylin and eosin-stained slides and laser-micro-dissected using a modified dissection instrument (Molecular Machines and Industries, Zurich, Switzerland). Collected tumor tissues were solubilized using liquid tissue (OncopPlex Diagnostics), according to the manufacturer’s instructions. The total protein concentration in each sample was measured by a micro bicinchoninic acid assay (Thermo Fisher Scientific Inc., Waltham, MA, USA). A mixture of stable, isotope-heavy, labeled synthetic peptides was added to the liquefied tumor samples as internal standards. All samples were analyzed in triplicate using a triple quadrupole mass spectrometer (TSQ Quantiva, Thermo Fisher Scientific Inc.) interfaced with a nanoACQUITY liquid chromatography system (Waters Corp., Milford, MA, USA). A chromatographic column packed with C18 resin with an inner diameter of 100 μm was used for peptide separation prior to mass spectrometry analysis. For protein quantitation, peak areas from each endogenous and heavy peptide were calculated and ratios were determined using PinPoint 1.3 (Thermo Fisher Scientific, Inc.). The limit of detection and quantification for ALK were 75 and 100 amol, respectively. The optimal quantification of peptides from seven other therapeutic target proteins was performed by trypsin digestion, and recombinant proteins specific for each target were mapped as described previously [33]. Protein levels are defined as detectable or not detectable for ALK, MET, FR-alpha, hENT1, RRM1, TUBB3, and ERCC1. Moreover, the protein level of XRCC1 is defined as high or low based on its median value. SRM was blinded to the clinical data and the FISH and IHC status.

### 4.4. Classification of Patients Treated with a Combination of Chemotherapeutic Agents into Protein Marker Groups

For each chemotherapeutic agent, we evaluated the correlation between the predictor protein levels obtained by SRM assays and clinical outcomes. For each patient, we divided the number of predictor proteins identified in their tumor sample by the total number of predictor proteins for the specific therapeutic agents they received. Next, we classified cases into five groups as follows: group 0, no predictive protein, group 1, <25%, group 2, 25–49%, group 3, 50–74%, and group 4, 75–100%.

### 4.5. Statistics

Statistical analysis was performed using GraphPad Prism 8 (GraphPad Software, San Diego, CA, USA) or SPSS software version 21.0 (IBM Corp., Armonk, NY, USA). The Fisher’s exact test was used to determine the association between the tissue fixation status and results of ALK testing. Kaplan-Meier survival curves were compared using the log-rank test. Results with *p* < 0.05 were considered statistically significant.

## 5. Conclusions

*ALK* FISH and IHC are currently approved for selecting therapy without orthogonal testing. However, every technique has limitations and confirmation of ALK status by using at least two techniques is recommended if maximal sensitivity and specificity for detecting *ALK* rearrangement is sought. In such a screening platform, we seldom encounter discordant cases and decision making for the patient is always difficult. Our preliminary result within limited cases shows that SRM may provide additional information for therapeutic decision making in patients with ambiguous ALK results.

## Figures and Tables

**Figure 1 cancers-13-02337-f001:**
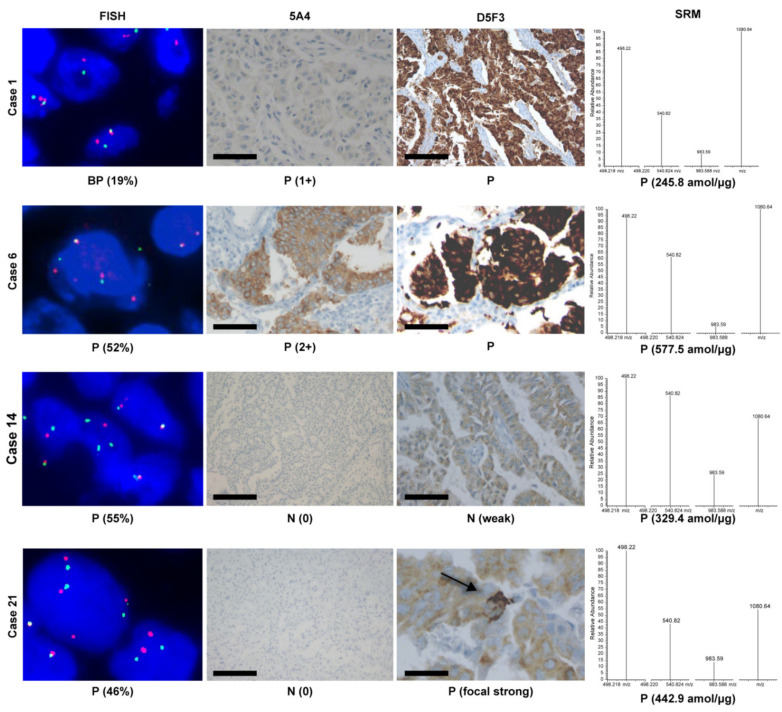
Illustration of additional immunohistochemistry with 5A4 and D5F3 antibodies, and SRM of cases previously discordant for anaplastic lymphoma kinase (*ALK*) FISH and ALK immunohistochemistry. Case number 1 is borderline-positive for *ALK* FISH, positive (1+) for 5A4, positive for D5F3, and positive for SRM. Case number 6 is positive for *ALK* FISH, positive (3+) for 5A4, positive for D5F3, and positive for SRM. Case number 14 is positive for *ALK* FISH, negative (0) for 5A4, negative (weak staining) for D5F3, and positive for SRM, and case number 21 is positive for *ALK* FISH, negative (0) for 5A4, positive (focal strong staining) for D5F3, and positive for SRM. Scale bars: 50 μm. Case numbers are summarized in Appendix A. FISH, fluorescent in situ hybridization. SRM, selected reaction monitoring. BP, borderline-positive. P, positive. N, negative.

**Figure 2 cancers-13-02337-f002:**
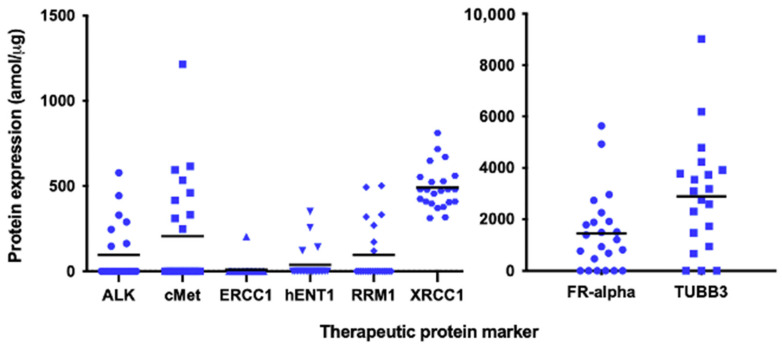
Scatter dot plot presents quantification of therapeutic protein expression by multiplexed, selected reaction monitoring assay. ALK, anaplastic lymphoma kinase. cMET, hepatocyte growth factor receptor. ERCC1, excision repair cross-complementation group 1. hENT1, human equilibrative nucleoside transporter 1. RRM1, ribonucleotide-diphosphate reductase M1. XRCC1, x-ray repair cross-complementing protein 1. FR-alpha, folate receptor alpha. TUBB3, tubulin beta-3 chain. Bar: mean.

**Figure 3 cancers-13-02337-f003:**
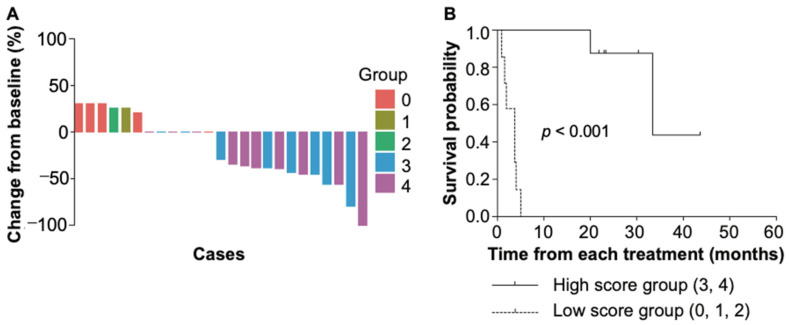
Therapeutic outcomes according to the protein expression quantified by selected reaction monitoring assay. (**A**) Best percent changes in tumor size by RECIST criteria based on the selected reaction monitoring marker groups. (**B**) Kaplan-Meier curve presents progression-free survival in cases, according to the selected reaction monitoring marker groups. RECIST, response evaluation criteria in solid tumors.

**Figure 4 cancers-13-02337-f004:**
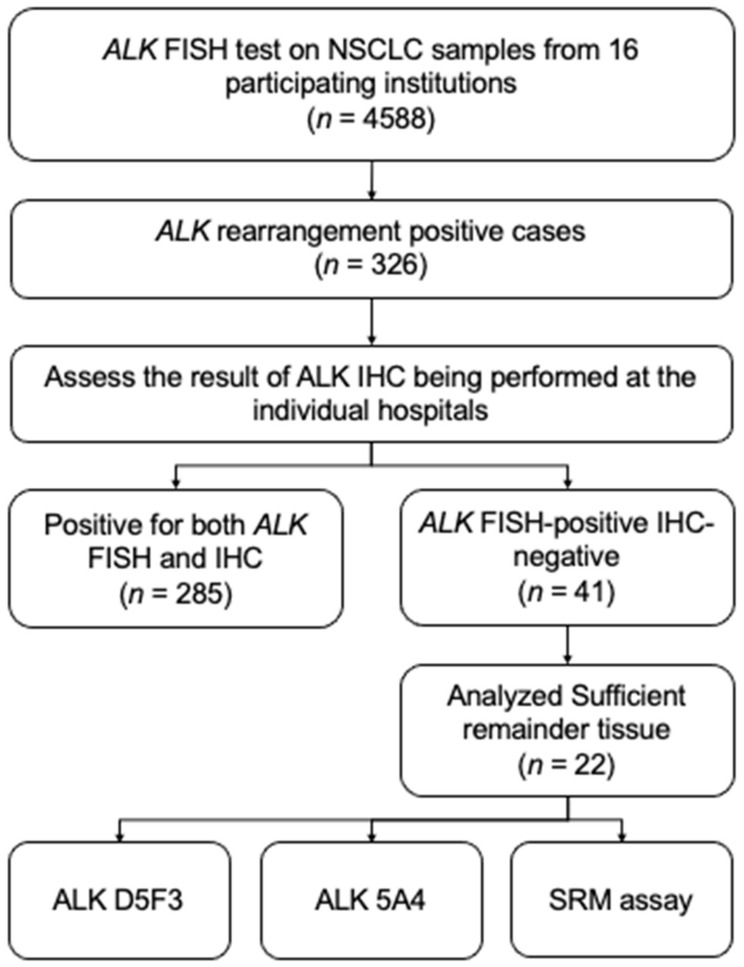
Schematic illustration of the study design and patient sample selection.

**Table 1 cancers-13-02337-t001:** Clinical characteristics of 22 discordant cases positive for *ALK* FISH and negative for ALK IHC.

Characteristics		*N* (%)
Age at Presentation (years)	Median	57
	Range	25–90
Sex	Male	13 (59.1)
	Female	9 (40.9)
Smoking	Non-Smoker	14 (63.6)
	Former/Current Smoker	8 (36.3)
Stage at Presentation	I–II	3 (13.6)
	III	3 (13.6)
	IV	16 (72.8)
Histology	Adenocarcinoma	21 (95.5)
	Adenosquamous Carcinoma	1 (4.5)
Specimen Type	Biopsy	15 (68.2)
	Lung	8 (36.4)
	Metastatic Site	7 (31.8)
	Lobectomy	7 (31.8)
Formalin Fixation	Direct Fixation (<30 min)	11 (50.0)
	Delayed Fixation (>30 min)	4 (18.2)
	Delayed Fixation (overnight)	7 (31.8)
ALK FISH	Positive (>20%)	8 (36.4)
	Borderline Positive (>15%, ≤20%)	14 (59.1)

*ALK*, anaplastic lymphoma kinase. FISH, fluorescent in situ hybridization. IHC, immunohistochemistry.

**Table 2 cancers-13-02337-t002:** Comparative analysis of *ALK* FISH, ALK IHC, and SRM of the 22 discordant cases.

*ALK* FISH		IHC				SRM	
	-	5A4 ^*^		D5F5 ^†^		ALK ^‡^	
	N	(+)	(−)	(+)	(−)	(+)	(−)
Positive (<20%)	8	2 (3+), 1 (2+)	5	4	5 ^§^	6 ^§^	2
Borderline (15–20%)	14	1 (1+)	13	1	13	1	13

* (+): 1+, weak cytoplasmic staining visible under 400× magnification, 2+, moderate cyto-plasmic staining clearly visible under 200× magnification, and 3+, strong cytoplasmic staining clearly visible under 40× magnification, (−): no staining. ^†^ (+) if a strong granular cytoplasmic brown staining was present in tumor cells (any percentage of positive tumor cells); (−) if there was no cytoplasmic staining or weak cytoplasmic staining. ^‡^ (+) if ALK protein was detected; (−) if not detected by SRM. ^§^ include two cases with weak negative staining. *ALK*, anaplastic lymphoma kinase. FISH, fluorescent in situ hybridization. IHC, immunohistochemistry. SRM, selected reaction monitoring.

**Table 3 cancers-13-02337-t003:** The association between tissue fixation and ALK test results.

ALK Detecting Methods	Direct Fixation (*n* = 11)	Delayed Fixation (*n* = 11)	*p*-Value
FISH	-	-	-
Positive	5	3	<0.05
Borderline	6	8	-
IHC (D5F3)	-	-	-
Positive	5	0	<0.05
Negative	6	11	-
SRM	-	-	-
Positive	5	2	<0.05
Negative	6	9	-

ALK, anaplastic lymphoma kinase. FISH, fluorescent in situ hybridization. IHC, immuno-histo-chemistry. SRM, selected reaction monitoring.

## Data Availability

Data is contained within the article or Appendix A. The data presented in this study are available in Appendix A.

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
