# Peer review of "Quantitative Multiplexed Proteomics Could Assist Therapeutic Decision Making in Non-Small Cell Lung Cancer Patients with Ambiguous ALK Test Results"

_cancers, 2021, doi:10.3390/cancers13102337_

Round 1

Reviewer 1 Report

Regarding the paper entitled” Therapeutic stratification of non-small cell lung cancers with ambiguous anaplastic lymphoma kinase test outcomes using quantitative multiplexed proteomics” the authors applied SRM to detect ALK and other protein biomarkers in patients with NSCLC with borderline or positive ALK FISH outcomes but negative ALK IHC results. They demonstrated that SRM could potentially guide therapeutic decision-making for patients with NSCLC with discordant ALK FISH and ALK IHC outcomes.

In my opinion the paper could be accepted after minor revision.

From the experimental point of view, the study is well planned, but really few samples arrived at the final step for proteomic quantification. In my opinion these few samples are not enough to propose an effective stratification of the patients. For this reason, the title of the paper should be revised emphasizing the fact that the proteomic analysis on few samples is not able to really give a stratification of the patients. Also the conclusion should be revised.

To enrich the dataset for a future continuation of the study, a full proteome profile is required, using high resolution mass spectrometry platforms.

Author Response

Response to reviewer1

Regarding the paper entitled” Therapeutic stratification of non-small cell lung cancers with ambiguous anaplastic lymphoma kinase test outcomes using quantitative multiplexed proteomics” the authors applied SRM to detect ALK and other protein biomarkers in patients with NSCLC with borderline or positive ALK FISH outcomes but negative ALK IHC results. They demonstrated that SRM could potentially guide therapeutic decision-making for patients with NSCLC with discordant ALK FISH and ALK IHC outcomes.

In my opinion the paper could be accepted after minor revision.

From the experimental point of view, the study is well planned, but really few samples arrived at the final step for proteomic quantification. In my opinion these few samples are not enough to propose an effective stratification of the patients. For this reason, the title of the paper should be revised emphasizing the fact that the proteomic analysis on few samples is not able to really give a stratification of the patients. Also the conclusion should be revised.

: Thank you for precious recommendations. We changed our corrections as italic character.

  • We revised our title as “Quantitative multiplexed proteomics could assist therapeutic decision making in non-small cell lung cancer patients with ambiguous ALK test results”
  • We revised our discussion as “our data suggest that SRM assays have superior sensitivity compared with IHC in poorly prepared tissue specimens and might provide useful supplementary information regarding the appropriateness of treatment with ALK inhibitors or other chemotherapies. Nonetheless, the present study is a retrospective study with a small sample size and not all cohort cases had enough tissue remaining for re-evaluation with IHC and SRM, which warrants further investigation to guide therapeutic decision-making for patients with NSCLC.”
  • We revised our conclusion as “Our preliminary results within limited cases show that SRM may provide an additional information for therapeutics decision making on patients with ambiguous ALK results.”

To enrich the dataset for a future continuation of the study, a full proteome profile is required, using high resolution mass spectrometry platforms.

: We included supplementary material regarding full proteome profiles and clinical information of our cohort.” 6. Supplementary Material Supplementary Materials:The following are available online at https://www.mdpi.com//1/1/0/s1, Table S1:  The results of selected reaction monitoring assay and evaluation of treatment responses.”

Reviewer 2 Report

The authors investigated the usefulness of SRM quantitative multiplexed proteomics in the ALK FISH-positive and IHC-negative lung cancers.

Since we occasionally encounter ambiguous cases at the diagnosis site, it is important to have additional methods for confirmation. It is likely that the TAT and cost of the test will be an important factor in determining whether it can be used in the clinical practice. It is recommended to describe the opinions of the authors on this point (factors influencing on clinical use: the amount of input material, TAT, and cost, et al..) in the discussion.

Author Response

Response to reviewer2

The authors investigated the usefulness of SRM quantitative multiplexed proteomics in the ALK FISH-positive and IHC-negative lung cancers.

Since we occasionally encounter ambiguous cases at the diagnosis site, it is important to have additional methods for confirmation. It is likely that the TAT and cost of the test will be an important factor in determining whether it can be used in the clinical practice. It is recommended to describe the opinions of the authors on this point (factors influencing on clinical use: the amount of input material, TAT, and cost, et al..) in the discussion.

Thank you for the precious comment. Our changes in manuscript are presented as italic character.

  • We revised details of SRM proteomic assay in material and method as:

“Briefly, a tissue section was sliced for hematoxylin and eosin staining, and a minimum of wo tissue sections at 10μm thickness for at least 1cm in diameter and consist of greater than 50% tumor or additional sections if tumor is smaller that 1cm were sliced onto the Director microdissection slides”

  • We inserted the description about the aspect of tissue requirement, turnaround time and cost in the discussion as:

In clinical setting, consideration of tissue requirement, cost and turnaround time in diagnostic platform are very essential {kerr2016precision}. The tissue requirement of SRM assay is similar with that of FISH and real time polymerase chain reaction, and even less than that of next generation sequencing. However, there are crucial limitations for the use of SRM in clinical setting. First, the cost of SRM is as high as or higher than that of next generation sequencing as SRM is currently not a companion diagnostic for ALK TKI and not reimbursed. Second, the availability of SRM is extremely limited that the turnaround time would be longer than any other commercially available method.
